# Food Security in Artisanal Mining Communities: An Exploration of Rural Markets in Northern Guinea

**DOI:** 10.3390/foods9040479

**Published:** 2020-04-10

**Authors:** Laetitia X. Zhang, Fatima Koroma, Mohammed Lamine Fofana, Alpha Oumar Barry, Sadio Diallo, Joseph Lamilé Songbono, Ronald Stokes-Walters, Rolf D. Klemm, Stella Nordhagen, Peter J. Winch

**Affiliations:** 1Department of International Health, Johns Hopkins Bloomberg School of Public Health, Baltimore, MD 21205, USA; fkoroma5@jhu.edu (F.K.); ronald.walters13@gmail.com (R.S.-W.); rklemm@hki.org (R.D.K.); 2Helen Keller International, New York, NY 10017, USA; mfofana@hki.org (M.L.F.); snordhagen@hki.org (S.N.); 3Julius Nyerere University of Kankan, Kankan, Guinea; aobarryunjk@gmail.com (A.O.B.); sadiomariamadiallo@gmail.com (S.D.); songbonoj@gmail.com (J.L.S.); 4Global Alliance for Improved Nutrition (GAIN), 1202 Geneva, Switzerland

**Keywords:** artisanal and small-scale mining, food security, diet, social environment, markets

## Abstract

The number of people engaged in artisanal and small-scale mining (ASM) has grown rapidly in the past twenty years, but they continue to be an understudied population experiencing high rates of malnutrition, poverty, and food insecurity. This paper explores how characteristics of markets that serve ASM populations facilitate and pose challenges to acquiring a nutritious and sustainable diet. The study sites included eight markets across four mining districts in the Kankan Region in the Republic of Guinea. Market descriptions to capture the structure of village markets, as well as twenty in-depth structured interviews with food vendors at mining site markets were conducted. We identified three forms of market organization based on location and distance from mining sites. Markets located close to mining sites offered fewer fruit and vegetable options, as well as a higher ratio of prepared food options as compared with markets located close to village centers. Vendors were highly responsive to customer needs. Food accessibility and utilization, rather than availability, are critical for food security in non-agricultural rural areas such as mining sites. Future market-based nutrition interventions need to consider the diverse market settings serving ASM communities and leverage the high vendor responsiveness to customer needs.

## 1. Introduction

Approximately 100 million people, including miners and their family members, are dependent on artisanal and small-scale mining (ASM) activities across 80 countries [1,2]. In contrast to industrial or large-scale mining, ASM is described as a labor-intensive activity conducted by individual miners or small enterprises with limited use of mechanical tools, limited capital investment, and low productivity [1,3]. Communities engaged in ASM tend to be impoverished and dependent on a livelihood with questionable sustainability. The World Bank, Washington, DC, USA describes ASM as “largely a poverty driven activity, typically practiced in the poorest and most remote rural areas of a country by a largely itinerant, poorly educated populace with few other employment alternatives” [1,2]. Moreover, their ranks are growing; the number of people directly engaged in ASM is estimated to have grown from 10 million in 1999 [4] to roughly 40 million people around the world [5].

ASM has consequences for not only sustainability but also human health. Similar to other rural communities in low- and middle-income countries, ASM communities are affected by various public health concerns, including malnutrition. Artisanal miners, who typically work with few or no mechanized tools, expend a considerable amount of energy in the process of digging into hard earth, breaking and hauling rocks, and washing and grinding them to find ore. While environmental and physical hazards related to ASM, such as mercury exposure and long working hours, have been relatively well documented [3], nutritional outcomes among ASM communities have not been as well researched [6]. This is despite the fact that food security is seen as an important motivator of involvement in ASM [3]. Most work that exists has focused on formal-sector miners in middle- or high-income countries [7], but some has considered those in lower-middle income countries such as Ghana [8,9]. Most of this work has focused on either nutritional status or diet and nutrient intake. For example, a study on the nutritional status of coal mine workers in India highlighted the double burden of undernutrition and overnutrition by indicating that 14.7% of workers had chronic nutritional deficiencies and 39.3% of workers struggled with overweight or obesity [10].

While many people are engaged in both subsistence farming and ASM [11], there are important characteristics of ASM communities that distinguish them from those engaged exclusively in subsistence farming. First, ASM communities consist of local populations as well as migrant populations. Migrant populations who settle temporarily in encampments close to mining sites in remote areas can face even greater barriers to accessing nutritious foods [3]. Due to the ability of miners to engage in ASM during the non-agricultural seasons or even year round, ASM communities are less prone to the seasonality inherent to farm-dependent livelihoods and are more dependent on using cash to purchase ready-to-eat foods [3,11]. Food environments can also change, as greater amounts of cash income and a population with limited access to home-grown food attract greater numbers of vendors and different types of foods. One study found that residents of Ghanaian artisanal and small-scale gold mining communities reported consuming more packaged and ready-to-eat foods, more sugar and fat, and less fruit and vegetables as compared with residents in surrounding rural areas who relied more on locally grown food items [6]. Through such shifts, ASM could have potentially negative impacts on diet quality and ultimately health. Given these key differences, it would be reasonable to dedicate special attention to studying the factors in the food environments that affect food security and nutritional outcomes in ASM communities. However, much of the existing food environment research focuses on nutritional disparities in urban areas, as well as high-income countries [12]. Additional research and interventions on food security in rural areas have focused on supporting subsistence farmers and improving their local food production [13,14].

The physical markets in which ASM communities interact are an integral part of their food environments and play a crucial role in their ability to acquire a safe, healthy diet. Most rural households in low-income countries rely extensively on markets to acquire nutrient-rich foods [15]. The three main components of food security, i.e., availability, accessibility, and utilization, are linked to market-related activities [16,17]. Food availability, which is the consistent presence of sufficient quantities of nutritious food, is affected by local production, as well as market purchases. Food accessibility, which is the presence of adequate resources for households to obtain nutritious foods, is affected by cost of foods, the markets’ hours of operation, and the distance to the closest market, particularly in the absence of public transportation. Food utilization, which is the proper biological use of food, is affected by hygiene including safety in food preparation, consumption, and storage. Turner et al. pointed out that markets in low- and middle-income countries are dynamic and complicated due to the widespread presence of informal, as well as formal, vendors [12]. Most existing market-related research in low-income countries has been centered on the relationship between market access and dietary diversity for households predominantly reliant on local food production [15,18]. Market access in these studies is defined as the distance between a farm household and the closest market where food can be sold or purchased. There has been limited research that has taken an in-depth look at the characteristics of markets that affect food availability, accessibility, and utilization for rural communities that rely heavily on food purchases [17].

In this study, we characterize eight markets that serve people engaged in artisanal and small-scale gold mining in the rural Upper region of the Republic of Guinea, West Africa. We aim to explore three questions. First, using the components of food security as a framework, how do characteristics of markets serving ASM communities present advantages and challenges for acquiring a safe and nutritious diet? Second, what characteristics could be leveraged to develop market-based nutrition interventions for these communities? Finally, what are the future research needs to support developing market-based approaches for improving the nutritional outcomes of ASM communities? We build on the understanding that individual dietary diversity is impacted by the food environment (including markets) in addition to other individual and household characteristics by discussing how components of food security are articulated in this understudied setting. We demonstrate that the market food environment is complex and dynamic, with miners in constant motion and vendors responding to the rapidly evolving needs of their customers.

## 2. Materials and Methods

### 2.1. Study Setting

The study, which was part of a larger research project, took place in two of the five prefectures located in the Kankan Region of the Republic of Guinea, the Kouroussa prefecture (population of 687,002) and the Siguiri prefecture (population of 268,630). While the Republic of Guinea is ethnically diverse, Kouroussa and Siguiri are predominantly composed of people of the Malinké ethnic group [19]. The primary language of communication is Malinké rather than the official language of French, as 80.9% of females and 67.9% of males in the Kankan Region have completed no level of formal education [19]. These areas represent the central area for artisanal gold mining in Guinea; almost the entire adult population in these districts is engaged directly or indirectly in mining and are representative of other mining communities in nearby countries [20]. The average mining site in the area hosts 3,720 people, 36% of whom are women and 37% children [20].

In Guinea, 74.1% of the population, or 9.4 million people, suffer from moderate or severe food insecurity, with rural populations three times as likely to be food insecure as compared with urban populations [21,22]. The 2018 Guinea Demographic and Health Survey (DHS) reported that of the children under the age of 5 in the Kankan Region, 30.5% are affected by stunting, 10.7% are affected by wasting, and 19.7% are underweight, giving the region the second highest rates of stunting, wasting, and insufficient weight in the country [19]. The DHS also reported significant disparities between rural regions as compared with urban regions for chronic malnutrition (33.8% rural vs. 21.7% urban) and insufficient weight (18.0% rural vs. 12.0% urban) [19,23]. In Kouroussa and Siguiri, the gold mining prefectures where the study took place, the prevalence of acute malnutrition in young children reached nearly 15%, which was considered a critical level [19]. Four villages in these prefectures were selected as research sites based on their engagement in mining activities, presence of a market, and accessibility by road during the rainy season. In this study, a market was defined as a physical setting where there was a cluster of more than one food vendor. There should be physical stalls, tables, or other constructs where vendors sell their food products, although ambulant vendors could also be present. For each of the four villages, we visited the market serving the residents of the village, as well as one of the informal markets located on the mining sites where residents worked. In total, eight distinct markets were included in this study. 

### 2.2. Data Collection

The larger study, results of which are reported elsewhere [24,25], included a series of scoping visits, key informant interviews, two stakeholder consultations (with government ministries, civil society agencies, and local miners’ associations), interviews and surveys of miners and their household members, and extended observations. The research was phased such that the initial results and the feedback from stakeholders were used to refine future topics for investigation. This paper focuses specifically on two distinct qualitative research activities from the final phase of the research, i.e., market descriptions and in-depth interviews with vendors. These qualitative research activities were designed to provide contextual information on the environment rather than be representative of the general population. Therefore, sample sizes were not planned to allow the generalization of results. The research activities in this study were originally approved by the Comité National d’Ethique pour la Recherche en Santé (CNERS) in Conakry, Republic of Guinea on 6 July 2018 (N: 080/CNERS/18). Approval was extended by the CNERS on 24 July 2019 (N: L-074/CNERS/19). 

Qualitative data collection took place in late July 2019. A market description guide was developed to capture the organizational structure and physical features of the markets serving each of the four villages. The guide consisted of a questionnaire to be completed with a market authority, observational questions, and guidelines for constructing a diagram of the market. Upon arrival at a market, interviewers met with local market authorities to present themselves and the project’s objectives and obtain permission to conduct the market description. Then, the interviewers conducted a brief structured interview with the market authority in order to gather information about the market’s history, organization, and operations. Next, the interviewers walked through the market to record observations of the market’s physical setting, including sketching a diagram. 

Following the completion of the market descriptions, structured interviews with food vendors were carried out at a total of four mining sites where residents of the selected villages worked. Given that food vendors were not present at every artisanal mining site, mining sites with food markets were located by asking village residents. Vendors were selected to participate in the interview if they primarily sold some sort of food product, if they spoke the local language, Malinké, or French, and if they consented to the interview. All but one of the interviews were conducted in Malinké using standard agreed-upon translations, with handwritten notes recorded in French. One interview was conducted and transcribed in French due to the language fluency of the vendor. Interview guides served to structure conversations between the interviewers and the vendors, focusing on topics including the vendor’s background, the types of foods sold, attitudes towards selling certain types of foods, and food hygiene. Interviews lasted approximately one hour. In total, twenty interviews with food vendors were completed across four mining site markets. Discussions with stakeholders, including the ministries of commerce and mining, revealed that little to no formal regulation or inspection of these markets takes place, particularly with regards to food safety and quality issues; as such, we did not include government representatives as subjects in the research.

### 2.3. Analysis

Field researchers from the University of Kankan, Kankan, Guinea and the Johns Hopkins University, Baltimore, Maryland, USA with training in qualitative methodology conducted preliminary analyses to summarize and organize data from the interviews and market descriptions. After data collection, all interviewer field notes, including vendor interview responses and market descriptions, were scanned and inputted into Microsoft Excel. The research team also took additional open-ended field notes even when they were not officially observing or interviewing. These included informal discussions and observations. The team noted the tone and attitudes of the respondents during data collection and met regularly during data transcription to generate relevant codes and themes. Coding was deductive (predefined) and inductive (emergent). The data analysis, in this paper, builds on the conceptual framework of the components of food security, as identified by Bashir and Schilizzi, as well as the dimensions of food access, as identified by Penchansky and Thomas [17,26]. Therefore, deductive codes were based on characteristics of markets and vendors in previous studies, the dimensions of food access (availability, accessibility, affordability, accommodation, and acceptability), and the components of foods security (accessibility, availability, and utilization) [16,17]. Researchers also generated inductive codes from the data, including attributes distinguishing various markets that serve mining communities, vendors’ motivations for selling certain food products, as well as concerns and benefits that interview participants associated with mining. Data were manually coded and categorized according to these major codes.

## 3. Results

### 3.1. Market Overview and Vendor Demographics

Through surveying eight markets within the selected mining sites and villages, we observed three ways in which markets were organized. Table 1 presents the basic characteristics that distinguish how we categorized the markets. Three markets were embedded in rural village centers, closer but not at mining sites (Type 1 in Table 1). One market was located along the periphery of a large village and served several smaller villages in the surrounding area (Type 2 in Table 1). Four markets were located directly at active mining sites and were relatively far from the villages where local miners resided (Type 3 in Table 1). In addition to differences in their proximity to the village and mining sites, the markets differed in terms of the days of the week that they operated and the number of vendors at the market. Artisanal mining site markets are comparatively small; if a mining site was served by a market, the market would consist of between two to ten food vendors. Our observations also revealed that these markets serve different groups of customers who vary in their involvement in mining activities (miners versus nonminers) as well as residence (the nearest village, nearby village, or far village/migrant). 

Basic demographic information of mining site vendors who participated in interviews is displayed in Table 2. Of the vendors interviewed, none identified the nearest village as their hometown. Eight of twenty vendors came from villages that were not involved in any mining activity. Eighteen of the vendors said that they found out about their mining site and vending location by word of mouth.

Figure 1, Figure 2 and Figure 3 depict the different ways in which the markets in the study were physically organized.

The diverging characteristics of these markets indicate variation in the availability, physical and temporal accessibility, and utilization of foods for customers. Table 3 summarizes market-related advantages and challenges to food security in ASM communities. 

The central village markets offered mining households a wide variety of nutritious food options that were both physically and temporally accessible. While the village periphery market offered an even wider variety of nutritious food options, the location and weekly nature of the market rendered it much less accessible. The mining site markets were easily accessible to miners working nearby, but the food options were far more limited. Limited health and sanitation infrastructure posed notable challenges towards supporting food utilization across all markets. The following subsections expand upon each of the food security components with regards to the markets in the study.

### 3.2. Food Availability

Table 4 summarizes the main food options available in the markets, grouped according to the food groups used to assess minimum dietary diversity (MDD) in young children [27]. It is clear that individuals who engage in ASM encounter considerable variability with regards to the consistent availability of diverse food options at the markets that serve their communities. While central village and village periphery markets offer a wide variety of foods encompassing all MDD food groups, mining site markets had a much smaller selection, offering limited or no unprocessed and uncooked food products including legumes and nuts, fruits and vegetables, eggs, and dairy.

Markets can offer a variety of food products, including prepared food items, such as cooked rice with meat and tomato sauce, grilled meat skewers, yogurt beverages, and small donuts (beignets), in addition to non-prepared staple foods and produce, such as sweet potato leaves, mangoes, eggs, and uncooked rice. The ratio of prepared to non-prepared food vendors ranged between the market types, with central village markets having the highest ratio of non-prepared foods and mining site markets having the highest ratio of prepared foods. Prepared food options offered at the markets tended to have a high proportion of staple grains, sugars, and fats with a lower proportion of meats and some vegetables.

### 3.3. Food Accessibility

Food accessibility encompasses market-related physical, temporal, and resource factors that affect customers’ ability to acquire nutritious foods. In terms of physical accessibility, a mining community would generally have direct access to either a market located in the center of its respective village (Type 1 in Table 1) or a market on the periphery of a neighboring village (Type 2 in Table 1). Not all mining sites have a market located at the site (Type 3 in Table 1). Rarely would all three types of markets be accessible to a given mining community. In terms of temporal accessibility, mining participants have much greater ease of access to foods if they are served by the daily central village or mining site markets as compared with the weekly village periphery markets. 

All markets have attributes that indicate a high responsiveness to the constraints of their respective customers and make their market experience more temporally accessible. Table 5 highlights key attributes of each market type that indicate a responsiveness to the preferences and needs of customers. The markets possess several features, depicted in Figure 1, Figure 2 and Figure 3, that are not directly food-related but are relevant, nevertheless. All of the markets in this study contained non-food vendors, offering products ranging from electronics, motorcycle parts, and medications, and non-food services, such as laundry services and motorcycle repair. These services render the markets more temporally accessible to customers by reducing the time customers need to travel elsewhere to handle these other non-food needs. 

Markets that are situated at the heart of large villages offer a wide range of non-prepared foods that customers can pick up to cook at home later. Their central location also offers their customers the flexibility to pick up ingredients while making trips to the health center or mosque rather than requiring customers to take extra time making separate trips to the market. However, this arrangement may not be helpful for the many women who are often actively involved in mining, given their limited time to prepare food [24,25].

The large market serving the nearest village in addition to several other smaller villages appears to offer a wide selection of nutritious foods, but it is only available weekly and requires some customers to travel a long distance, roughly 18 km, by foot or by motorcycle if available. Access is further limited by the conditions of the roads, which are narrow and unpaved, contain deep potholes, and experience frequent flooding during the rainy season. The large weekly market adapts to entice customers traveling far with a combination of unprepared foods to bring home to cook as well as hot, prepared foods to eat during the market trip. This market is situated right by the side of the road with gas pumps, motorcycle repair services, and a parking lot, making it an appealing rest stop for traveling gold miners or other people moving through the region. Vendors advertising their product prices through speakers and megaphones were helpful to customers trying to find certain products amid a bustling and densely packed marketplace.

The markets situated at the periphery of mining sites were inherently most specifically tailored to meet the needs of customers engaged in mining activities. A mining site market offers an easily accessible opportunity for miners to grab a meal during their workday, but the selection of nutritious options is much more limited than at a larger market. Interviews with mining site market vendors revealed how they take client preferences and needs into consideration in determining where their stall is situated, what products they sell, and how they choose to conduct their business. All vendors interviewed sold prepared food items, such as fried cakes and sandwiches; rice with a sauce was the most popular item sold (10 out of 20 vendors). None of the vendors sold unprepared staple foods or produce. When asked why they sold the items they did, vendors’ main reasons included client preferences (e.g., “it’s what customers like”, eight vendors), habit (e.g., “I have always sold this” or “this is what I know to make”, eight vendors), and convenience (e.g., “it’s easy to sell these items while looking after children”, three vendors,).

### 3.4. Food Utilization

Food utilization is influenced by hygiene practices not only within households but also by food vendors at markets. All the vendors interviewed at the mining sites noted that they are mindful of hygiene in their food preparation and service. The top reasons vendors cited for caring about hygiene were for attracting or retaining customers (18 vendors), to prevent illness (10 vendors), and because they or their children eat the same food (eight vendors). Note that vendors could cite multiple reasons for choosing to sell certain items or for paying attention to food hygiene. This finding reinforces a finding of the larger research program that local customers see “healthy foods” as being those that are not dirty, expired, or spoiled [25]. Common methods that vendors employ to preserve hygiene include covering prepared foods with a cloth, washing plates, wiping down the eating area, and wearing clean clothing. Similar hygiene practices were observed at central village markets and the village periphery market.

Despite their knowledge of and attention to food hygiene, the vendors at mining site markets face major constraints in maintaining food hygiene in practice. Due to their remote locations, mining site markets had limited access to clean water. While several of the village markets had limited access to privately owned electric generators, the mining site markets universally lacked access to electricity, posing considerable constraints on the vendors’ ability to preserve the freshness of ingredients and food products. Several of the vendors expressed that they chose to sell specific food items because they wanted to avoid bringing back perishable leftovers at the end of the day. Furthermore, the absence of a waste disposal system at all markets posed potential health concerns due to the accumulation of food and plastic waste on the market grounds, as well as smoke fumes from burning waste.

## 4. Discussion

In order to understand food choices and nutrition in rural sub-Saharan Africa, the diversity of food environments and markets serving communities that do not strictly rely on local food production needs to be recognized. In this study, we captured a variety of characteristics distinguishing the dynamic marketplaces that serve artisanal gold-mining communities in rural Upper Guinea. Characteristics such as proximity to villages and mining sites, types of products sold, and organization can have implications for the ability of community members to obtain a healthy, diverse diet and to do so in a sustainable way. The primary advantages of the markets for helping customers achieve food security were the availability of large quantities and a diverse array of nutritious foods at most village markets. However, there were also serious challenges associated with achieving food security based on existing market characteristics. The main challenges were related to the temporal and physical accessibility of the market food options, as well as environmental constraints to food hygiene practices, particularly at mining site markets. 

One market characteristic that stood out as both advantageous, as well as obstructive to achieving food security, was the high level of responsiveness of vendors to customer needs. This concept, emphasizing the dynamic nature of food environments in this setting, is similar to the understudied feature of food environments known as accommodation [28]. The eight markets in this study demonstrated a responsiveness to their gold-mining clients’ needs in a variety of ways, with the four mining site markets demonstrating the most specialized response to the needs of their customers engaged in mining. The ways in which markets respond to their customers’ needs impact the nutritional content of the foods sold and the attractiveness of the markets to potential customers. In this manner, customers may be willing to acquire food from a given market even if other factors, such as location and hours of operation, are not ideal. The widespread presence of seating areas in mining site markets, a feature that was absent from village center markets and the large village periphery market, is one way in which vendors sought to attract customers by offering them a place to rest throughout their day of labor-intensive mining activities. 

On the one hand, the high level of accommodation seems to pose an obstacle towards helping miners access nutritious and sustainable food options. For instance, exclusively selling packaged and prepared foods at the mining site markets is another way in which vendors respond to the needs of busy customers engaged in mining, but given the relatively low levels of fruits, vegetables, and, in some cases, protein sources in these meals, these convenient meals do not necessarily represent healthy options at mining sites. Migrant miners who live at encampments near the mining sites and almost exclusively have access to a mining site market are particularly dependent on these meals. Our observations reinforce a finding from a study in Ghana that high migration areas appear to have an increase in food expenditures on less nutritious foods such as premade foods, sugar, and beverages [29]. Moreover, selling commercially packaged foods such as biscuits, sodas, candies, and water sachets could lead to high levels of waste, which cannot be safely processed locally in these local areas. This can decrease the environment sustainability of the local food market. 

On the other hand, the high level of accommodation also has the potential to be leveraged to improve food security through future market-based interventions. With regards to food utilization, customers have the ability to influence the actions of market vendors to better meet their needs. For instance, vendor interviews revealed that the ability to attract and retain customers was the main reason for paying attention to hygiene in food preparation and service. Health codes, government regulations, or any other established hygiene standards that commonly exist in high-income settings were not mentioned at all [30,31,32]. Indeed, stakeholder consultations carried out during the research planning phase revealed that government oversight of these markets included food safety issues. While hygiene has been cited as an important element of food utilization, research has tended to focus on hygiene practices at the household level [17]. Our findings suggest that customers also place value on and have the capacity to influence hygiene practices at markets.

The responsiveness to customers’ needs also applies to improving food security in relation to food accessibility. There has been some research on how ASM can complement subsistence farming and improve access to foods in remote areas. For example, Cartier and Bürge suggested that mining areas increase the demand for consumer goods, which can be satisfied by local agricultural production and contribute to market decentralization, or the movement of markets from traditional urban centers to rural peripheries [33]. The makeshift construction of the mining site vendor stalls, fashioned from sticks, leaves, and branches, reflects that mining sites sporadically emerge and close and food vendors appear and disappear in tandem. Still, due to the high proportion of staple-heavy, low-nutrient-dense prepared foods sold at mining site markets in comparison to village markets, ASM activities alone do not seem to attract non-staple food vendors and benefit from being complemented by government or donor support to improve access to nutritious foods in remote areas [33].

This study had certain weaknesses, including the small sample size of markets due to the limited number of markets in operation during the rainy season, when data collection took place. This study is not intended to be a comprehensive typology of Guinean or ASM markets, but rather an attempt at capturing the diverse characteristics of markets serving ASM communities. Additionally, although the study sought out native speakers to conduct the interviewers and took care to agree upon suitable translations of questions between Malinké and French in advance, it is possible that some richness of the participants’ responses was lost in translation. This study’s strengths included the combination of several qualitative research methods to elucidate meaningful patterns. For instance, the vendor interviews, as well as discussions with market authorities, allowed for the in-depth exploration of patterns that emerged during market observations. This study also relied heavily on locally-based research team members, which enabled us to better capture cultural nuances. 

## 5. Conclusions

Through studying eight markets serving mining communities in the Kankan Region of Guinea, our study provides preliminary insight into how components of food security are articulated in this understudied setting. Our findings suggest that food accessibility and utilization, rather than availability, are critical for food security in non-agricultural rural areas such as mining sites. The complexity of the markets studied, here, has implications for the design and implementation of future market-based interventions to improve nutritional outcomes within non-agricultural rural communities. First, as noted by Turner et al. [12], we emphasize the need to characterize and classify markets that serve these communities in order to better understand factors driving food choices and food acquisition. Terms frequently used in existing food security literature such as “supermarket” and “corner store” are not meaningful in low-income countries [12]. The presence of mining site markets suggests the need to define meaningful food environment metrics that apply to informal markets. Additionally, given that mining participants face a diverse range of challenges in achieving dietary diversity, efforts should be made to continue exploring the interface between household factors and local markets’ characteristics in order to determine priority factors for market-based dietary interventions in different communities. Furthermore, future interventions should be designed with the understanding that vendors are strongly influenced by the perceived needs of their customers. Markets can be leveraged to disseminate information about nutrient-rich foods and diets, and vendors may be more receptive to selling certain nutritious food options if those options also meet the specific needs of their clients. This could be accomplished through targeted consumer behavior change communication to improve knowledge of nutrition and dietary diversity [18]. Market-based interventions that seek to improve dietary diversity, should take a multifaceted approach that seeks to change behavior on both the vendor and client ends, and vendors should be encouraged to offer feedback on the feasibility of interventions. 

## Figures and Tables

**Figure 1 foods-09-00479-f001:**
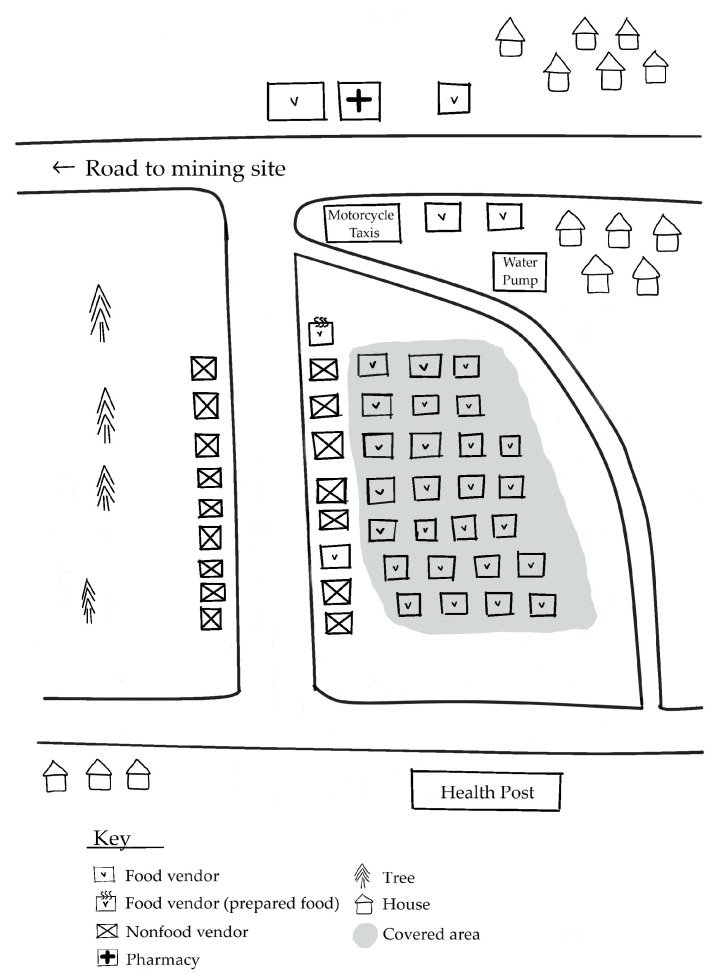
Map of market located in village center. Typical of a market where most food vendors sell unprepared foods.

**Figure 2 foods-09-00479-f002:**
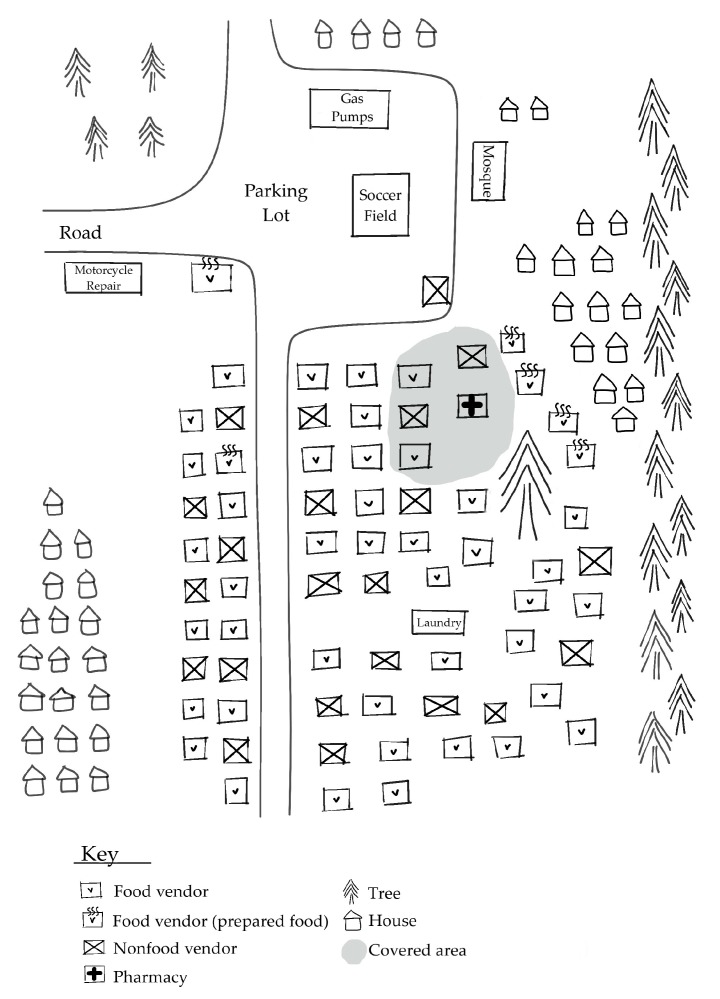
Map of large market on the periphery of a village. Combination of prepared and unprepared food items. High volume of vendors.

**Figure 3 foods-09-00479-f003:**
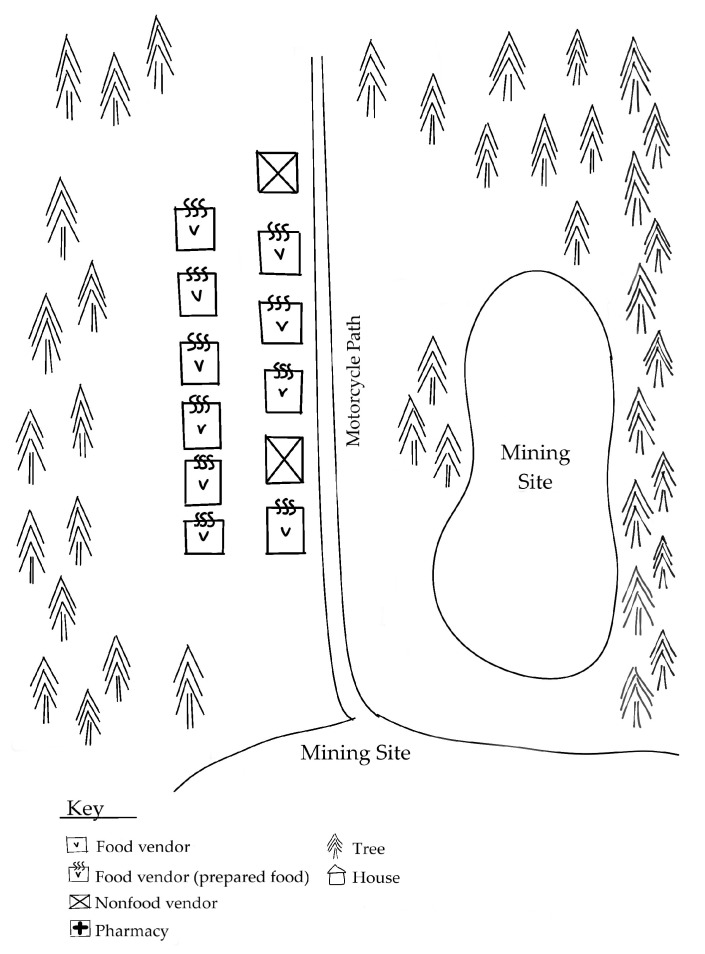
Map of mining site market. Most vendors sell prepared foods. Large stalls include seating areas for customers.

**Table 1 foods-09-00479-t001:** Three types of markets serving mining communities in rural Upper Guinea.

	Type 1 (*n* = 3)	Type 2 (*n* = 1)	Type 3 (*n* = 4)
Location	Village center, farther from mining site	Edge of large, permanent village, 18 km from smaller villages, closer to mining site	Close to, or located directly on, mining site
Frequency with which market is open	Daily	Weekly	Daily
Approximate number of vendors	50	80	10
Customers served by the market	Mining and nonmining residents of village	Mining and nonmining residents of closest village; residents of smaller, nearby villages	Local mining participants; migrant mining participants residing in nearby encampment

Total number of markets = 8. *n* indicates the number of markets in each category.

**Table 2 foods-09-00479-t002:** Descriptive characteristics of vendors interviewed at mining site markets.

Demographic Characteristic	-
Age * in years, mean (SD)	27 (11)
Gender	*n* (%)
Female	20 (100)
Level of Education	*n* (%)
None	7 (35)
Some primary school	9 (45)
Some secondary school	3 (15)
College degree	1 (5)

Total sample population = 20. *n* indicates the number of vendors in each category. * Four of the twenty vendors indicated that they did not know their age. Vendors at village center and village periphery markets were not asked to participate in interviews.

**Table 3 foods-09-00479-t003:** Summary of market-related advantages and challenges to ASM food security, organized by food security component.

Food Security Component	Definition *	Advantages	Challenges
Food availability	Sufficient quantities of food are consistently available to all individuals.	Large quantity and variety of foods at village markets; all food groups represented in central village and village periphery markets	Foods locally sourced, leading to seasonal variation in availability of certain foods; limited availability of some food groups (legumes and nuts, fruits and vegetables) at mining site markets
Food accessibility	All individuals have adequate resources to obtain appropriate foods for a nutritious diet	Ease of physical access to central village and mining site markets; daily operation of central village and mining site markets; all markets demonstrate responsiveness to customer needs	Difficulty in accessing village periphery markets due to distance and weekly operation; limited financial accessibility of non-staple foods
Food utilization	Proper biological use of food, includes diet providing sufficient energy and essential nutrients, potable water, and adequate sanitation	Vendors express conscientiousness towards food hygiene	No waste disposal system; limited electricity for food preservation (absent at mining site markets); limited clean water access, particularly at mining site markets

* Definitions adapted from Bashir et al. [17].

**Table 4 foods-09-00479-t004:** Types of foods available by food group and market.

Food Group	1: Markets Located in the Center of Permanent Village (*n* = 3)	2: Market Located on the Periphery of Permanent Village, Serving Surrounding Villages (*n* = 1)	3: Markets Located at the Mining Site (*n* = 4)
Grains, roots, and tubers	Maize (flour and kernels), rice, sorghum, fonio, dried pasta, bread, yams, plantains, white sweet potatoes, potatoes	Maize (kernels), rice, millet, wheat flour, fonio, dried pasta, bread, cassava, white sweet potatoes, potatoes	Cooked rice *, attieke (fermented and cooked cassava) *, fried white sweet potatoes *, cooked spaghetti *, bread
Legumes and nuts	Cowpeas, peanuts	Beans, peanuts	
Dairy products	Fresh milk, powdered milk	Fresh milk, powdered milk	Powdered milk *^,^^, homemade yogurt drink *^,^^
Flesh foods	Fresh fish, dried fish, sardines, beef, chicken (alive and local, frozen and imported), grilled goat meat *, live goats	Fresh fish, dried fish, sardines, beef, chicken (alive and local, frozen and imported)	Grilled meat skewers *, sauce containing chunks of chicken, fish, or beef *
Eggs	Fresh eggs	Fresh eggs	Omelet *^,^^
Vitamin A rich fruits and vegetables	Sweet potato leaves, tomatoes	Sweet potato leaves, mangoes	Sauce containing sweet potato leaves *, sauce containing tomatoes *
Other fruits and vegetables	Onions, garlic, eggplants, African eggplants, cucumbers, bananas, avocados	Onions, eggplants, African eggplants, cucumbers, bananas, avocados	Fried plantains *

*n* indicates the number of markets in each category. * Indicates prepared food item ^^^ Indicates that item was only seen at one vendor across four mining sites. Limited availability of certain food groups (legumes and nuts, dairy, eggs, and fruits and vegetables) at mining site markets.

**Table 5 foods-09-00479-t005:** Comparison table showing how markets are organized to respond to customer needs and improve accessibility of products.

Market Feature	1: Markets Located in the Center of Permanent Village (*n* = 3)	2: Market Located on the Periphery of Permanent Village, Serving Surrounding Villages (*n* = 1)	3: Markets Located at the Mining Site (*n* = 4)
Ratio of staple to prepared foods	Almost all unprepared/raw	About even	Almost all prepared
Non-food products and other services	Clothing, shoes, electronics, kitchenware	Clothing, shoes, electronics, medications, motorcycle parts, gold, laundry service	Clothing, toys for children, knives, toothbrushes, traveling dentists
Seating areas	None	None	Present with most food vendors
Construction of stalls	Wooden stalls; large, connected tarp covering	Individual stalls; individual umbrella, metal, or tarp covering	Individual stalls constructed from branches; individual covering made from sticks or tarp
Presence of ambulatory vendors	No	Yes	Yes
Ambience and advertising	A lot of bustle and chatter. Vendors advertising prices of merchandise through audio recordings.	Lively, lots of chatter. At times, music playing. Vendors broadcasting prices by megaphone.	Quiet chatter. No music or audio recordings.
Nearby landmarks	Close to health post and primary school	Close to parking lot, mosque, gas pumps, row of houses, small soccer field, and motorcycle repair station	Mining site

*n* indicates the number of markets in each category.

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
