# Peer review of "Food Security in Artisanal Mining Communities: An Exploration of Rural Markets in Northern Guinea"

_foods, 2020, doi:10.3390/foods9040479_

Round 1
Reviewer 1 Report
The study is very interesting for reader. The topic is very specific toward the sustainability of local market in a specific area. The authors present a strong motivation for the study according to the scientific approach. However, it’s recommended to integrate some parts of the paper:
- I recommend underlining in the abstract the contribution of the paper.
- It is suggested to move the lines 86-97 to materials and methods paragraph, because starting from line 86 the research hypotheses of the paper can be highlighted
- In the section “Materials and methods” I recommend adding a separate paragraph dedicated to the detailed description of the study area.
- What kind of analysis has been done? It is not clear (lines 137-141)
- It is recommended to digitize visual content for greater graphic consistency (Fig 1,2,3)
- I suggest you to end the discussion paragraph at line 339
- From lines 340 it’s advisable to insert a paragraph "Conclusions" separate from the discussions.
- References are adequate and coherent. However,(for example in the case of dated articles) it is recommended to update it.
Author Response
R1_Comment01. The study is very interesting for reader. The topic is very specific toward the sustainability of local market in a specific area. The authors present a strong motivation for the study according to the scientific approach.
R1_Comment02. I recommend underlining in the abstract the contribution of the paper.
- Thank you for this suggestion. The contribution of the paper has been underlined in the revised abstract (lines 23-26).
R1_Comment03. It is suggested to move the lines 86-97 to materials and methods paragraph, because starting from line 86 the research hypotheses of the paper can be highlighted.
- Thank you for the suggestion. However, we have decided to leave lines 86-97 as the final paragraph of the Introduction section, as they introduce the research questions around which the rest of the paper is organized.
R1_Comment04. In the section “Materials and methods” I recommend adding a separate paragraph dedicated to the detailed description of the study area.
- That is a good point. We have added in a separate subsection at the beginning of the “Materials and methods” section dedicating to describing the study area (lines 105-132).
R1_Comment05. What kind of analysis has been done? It is not clear (lines 137-141)
- We have added in a separate subsection titled “Analysis” to the “Materials and Methods section” that elaborates upon the type of analysis being done (lines 170-188).
- Field researchers from University of Kankan and Johns Hopkins University with training in qualitative methodology conducted all data collection in Malinke and French using semi-structured interviews and market descriptions. Field researchers recorded all data collection events and conducted preliminary analyses to summarize and organize data from interviews and market descriptions. The research team also took additional open-ended field notes even when they were not officially observing or interviewing. These included informal discussions and observations. The team noted the tone and attitudes of the respondents during data collection and met regularly during data transcription to generate relevant codes and themes.
- The research team met regularly during data collection. Deductive codes were based on characteristics of markets and vendors in previous studies, the dimensions of food access (availability, accessibility, affordability, accommodation, acceptability), and the components of foods security (accessibility, availability, and utilization). We also generated inductive codes from the data, including attributes distinguishing various markets that serve mining communities, vendors’ motivations for selling certain food products as well as concerns and benefits that interview participants associated with mining. Data were manually coded and categorized according to these major codes.
R1_Comment06. It is recommended to digitize visual content for greater graphic consistency (Fig 1,2,3)
- Text of the images has been digitized for greater consistency (lines 216-224).
R1_Comment07. I suggest you to end the discussion paragraph at line 339
- Thank you for this suggestion. We have added a conclusion section to the paper (lines 409-432).
R1_Comment08. From lines 340 it’s advisable to insert a paragraph "Conclusions" separate from the discussions.
- Same note as above (lines 409-432).
R1_Comment09. References are adequate and coherent. However,(for example in the case of dated articles) it is recommended to update it.
Thank you for your recommendation. The distribution of date of publication of sources cited is: 1981 (1), 1999 (1), 2002-2009 (5), 2010-2020 (27). This is reasonable. The 1981 article by Penchansky provides basic definitions used in the field. The 1999 ILO report was the largest study ever conducted on artisanal mining. We see no need to exclude these references.
Reviewer 2 Report
- line 76 "distance to the market". Because access to remotely located markets carn be offset through some form of public transport, similarly closer markets that have no public access can present greater accessibility challenge, could this sentence please be reworded to better explain. If no public transports options (then please state) in which case distance is the determinant.
- Material and methods. Noting research was undertake "in four villages" [line 99]; and that "four research sites were selected based on their level of mining activity, accessibility and market presence" [line 108]; and "interviews completed with food vendors in a total across four mining site markets" [136], its not clear exactly how many markets were assessed in which villages and their relationship to mining sites (i.e. were there multiple mine site per village or similarly multiple markets per village]. I note a total of eight markets were surveyed [line 57 i.e. n=3+n=1+ N=4]. Suggest a small rewording to line 136 to clarify.
- Table 3, and Table 4. The formatting of information in column 2-4 needs to be amended to avoid confusion. For example Table 3 - column 3 "Ease of physical access to central......" Is this text linked to the row related to food availability or the row related to food accessibility?
- Replace "trash" with "waste" re table 3 and line 277.
- Table 3. "fried plantains" is not a grain, root or tuber.
- Table 3. last row. Some products are listed in plural (i.e.. onions) where as other products are not. Best to be consistent.
- Insert line space after table 3 and table 4 footnoting, before the next paragraph.
- Line 239 "long distance". Can you please qualify what you consider as long distance.
Author Response
R2_Comment01. line 76 "distance to the market". Because access to remotely located markets carn be offset through some form of public transport, similarly closer markets that have no public access can present greater accessibility challenge, could this sentence please be reworded to better explain. If no public transports options (then please state) in which case distance is the determinant.
- Thanks for pointing this out. In the sites that we studied, there is no access to public transportation. Select individuals who have access to a personal motorcycle will use that. Everyone else will walk. Mobility is further limited by difficulties with the terrain. For instance, deep potholes as well as frequent flooding of the roads are common during the rainy season. This has been added to the manuscript in line 81 as well as lines 295-297.
R2_Comment02. Material and methods. Noting research was undertake "in four villages" [line 99]; and that "four research sites were selected based on their level of mining activity, accessibility and market presence" [line 108]; and "interviews completed with food vendors in a total across four mining site markets" [136], its not clear exactly how many markets were assessed in which villages and their relationship to mining sites (i.e. were there multiple mine site per village or similarly multiple markets per village]. I note a total of eight markets were surveyed [line 57 i.e. n=3+n=1+ N=4]. Suggest a small rewording to line 136 to clarify.
- You are correct in that we collected data at eight markets, and I agree with your suggestion that we ought to clarify the number of markets. Further clarification has been included in lines 125-126 and line 129-132.
R2_Comment03. Table 3, and Table 4. The formatting of information in column 2-4 needs to be amended to avoid confusion. For example Table 3 - column 3 "Ease of physical access to central......" Is this text linked to the row related to food availability or the row related to food accessibility?
- Good catch, I can see how this could be confusing. The formatting of the tables has been adjusted so that the rows are aligned appropriately.
R2_Comment04. Replace "trash" with "waste" re table 3 and line 277.
- These changes have been made in Table 3 and line 334.
R2_Comment05. Table 3. "fried plantains" is not a grain, root or tuber.
- Thanks for catching our error. Table 3 has been corrected by moving “fried plantains” to the “other fruits and vegetables” section.
R2_Comment06. Table 3. last row. Some products are listed in plural (i.e.. onions) whereas other products are not. Best to be consistent.
- Thanks for pointing this out. The products are now all listed in plural for consistency in the last row of Table 3.
R2_Comment07. Insert line space after table 3 and table 4 footnoting, before the next paragraph.
- A line space has been inserted after the table 3 and table 4 footnotes.
R2_Comment08. Line 239 "long distance". Can you please qualify what you consider as long distance.
- Residents of a small village, including people engaged in mining, resided roughly 18km away from the weekly market located on the periphery of a larger village. Residents of the small village would travel by foot or share a ride on a motorcycle. This has been added to the paper in lines 295-297.
Reviewer 3 Report
I think that this is a really interesting bit of research in a under-researched area (food security in informal mining areas) with an fairly novel quantitative approach. I strongly feel that it should be published as it can encourage others to shed light on these issues further.
Notwithstanding, I have some concerns and/or suggestions for the authors to consider.
Energy and nutrition intake. Are there any special needs for miners? Don't they work particularly hard and expend a great deal of energy? Informal miners probably do not have equipment to move large amounts of material and so have to do this by hand. This issue is rather glossed over.
Much food in informal settlements is delivered by mobile food vendors. There seems to be a missing step in the method where they should have done a stakeholder analysis. The focus is on stationary food vendors. No interviews were done with consumers or local farmers. Local governments role in managing food vending areas is not mentioned and no state actors were interviewed. I think the method is defficient in not mapping the actors (e.g., with value chain analysis) or having some interviews with a wider set of relevant actors. It is too late now to go back, but the narrowness of the sample and needs to be explained and possibly a wider study method suggested in future which addresses this issue.
A study like this where data is collected from potentially vulnerable individuals should have been approved by the relevant research ethics committee at one of the authors institutions and this should be mentioned. If there was no ethics approval for the method, then how do I know that it was approved and done to a known standard. Not highlighting the ethics approval threatens the reputation of the journal.
I assume that in an informal mining settlement many different cultures are brought together so a lot of different food traditions might be present. Not sure if this is covered or even asked.
p2 line 62. Are there no positive outcomes? What about opportunities for women to enter the food service business. it would be good to posit some benefits as well as many down-side issues.
p2 line 64. It is essential. Maybe use a more passive voice?
p2 para ending line 68. I have the feeling that formal mining has received a lot of research interest - for example in South Africa and Latin America. Some more reference to this would have been helpful.
p2, line 82. This is a very narrow definition of market research, so you need to specify that it is specific to this research.
p2, from line 90. How were these questions selected?
I feel that this section would have benefited from a conceptual framework, even if only a simple one.
p3, line 96. You may have demonstrated complexity, but was this in answer to a research question, or just an incidental outcome. Vendors responding to customers in a competitive environment is not a particularly exciting or surprising finding.
p3, line 117 on. This is where ethical clearance needs to be mentioned.
p4. When thinking about the method, I was left wondering why a counterfactual had not been included - ie., a community with no mining activity going on. This would have given a control for the other cases.
Figs 1 - 3. The key is the same for all pictures, but not everything appears in all figures - for example covered area. I think this needs to be adjusted. Do these work if smaller? They take a lot of space.
Table 3. It is not clear how this table was made. Is this the opinion of the authors or was a process of some kind? It contains some value judgements e.g., about food hygiene. I think the source needs to be authors opinion.
Throughout the piece ethnography seems to be absent. What about traditional and cultural practices? Beliefs and taboos related to food etc? Many food and dishes in a country such as Guinea come associated with beliefs and practices related to culture.
There is no mention of forest foods or wild crafted foods or herbal remedies in table 4. I do not believe that a mine in the middle of a forest does not have somebody selling stuff that has been wild harvested. I also do not believe that none of these markets had no green leafy vegetables or tomatoes. Because none of the supply chains/ value chain have been unpacked, we do not know where all this food really came from or which other actors are 'hidden' from view. We also discover very little about the vendors themselves, which is a bit of a shame.
p11, line 266. Some terms are used very vaguely in this paper. An example in this line is 'spoiled'. What does this mean. Often food that is damaged or starting to physically deteriorate still has a value, particularly for the poorest.
The piece is strangely silent about the role of any government actors in managing these vending sites. Maybe there is no role.
p12, line 296 onwards. I don't see how you can detach the food environment from the other needs of miners. Other studies show that mines attract alcohol sellers, prostitutes, drugs, etc etc. Miners need equipment as well as food.
The idea of accommodation in introduced here, but there is no mention of this in the introduction. If you were going to test whether accommodation is present, then this should have been a hypothesis and it would have improved the piece.
p12, line 334. I don't think twigs is the right term - should be stick or poles.
line 337. I find the recommendation about non-staple food vendors a bit of a stretch from the data.
p13, line 352. They correctly cite Turners advice, but I see no evidence that they followed it, ie., characterised or classified markets, at least not adequately.
line 364. I am very sceptical about this finding, particularly as they have not included analysis of the various different actors views on the issue because they did not interview policy makers, donors or government actors - so this is just opinion.
Author Response
R3_Comment01. I think that this is a really interesting bit of research in a under-researched area (food security in informal mining areas) with an fairly novel quantitative approach. I strongly feel that it should be published as it can encourage others to shed light on these issues further. Notwithstanding, I have some concerns and/or suggestions for the authors to consider.
R3_Comment02. Energy and nutrition intake. Are there any special needs for miners? Don't they work particularly hard and expend a great deal of energy? Informal miners probably do not have equipment to move large amounts of material and so have to do this by hand. This issue is rather glossed over.
- This is true and demonstrates yet another reason to focus more research attention on the topic of nutrition in these areas – though it must be kept in mind that the main alternative livelihoods in these rural areas is smallholder agriculture, which is also physically demanding. Assessing the actual energy needs or nutrient intake of informal miners is beyond the scope of this study, but we have noted this point in the manuscript (p. 1-2, lines 43-45).
R3_Comment03. Much food in informal settlements is delivered by mobile food vendors. There seems to be a missing step in the method where they should have done a stakeholder analysis. The focus is on stationary food vendors. No interviews were done with consumers or local farmers. Local governments role in managing food vending areas is not mentioned and no state actors were interviewed. I think the method is deficient in not mapping the actors (e.g., with value chain analysis) or having some interviews with a wider set of relevant actors. It is too late now to go back, but the narrowness of the sample and needs to be explained and possibly a wider study method suggested in future which addresses this issue.
- Thank you for pointing this out. We agree that it is essential to include the views of other stakeholders, and this piece of research is only one part of a larger study that examined multiple other perspectives. Extensive work with miner-consumers (many of whom are also farmers) was undertaken, including semi-structured interviews, observations, and a household survey. Results from analyzing this data will be reported separately as it examines slightly different research questions and due to length limitations (these forthcoming publications are noted in the manuscript on p. 3, line 134), however, the insights from that work do inform the focus of this paper as well as the conclusions it draws.
- The overall study began with site scoping visits and key informant interviews (with government, civil society groups, and local miners), which were used to identify key stakeholders. We then held two stakeholder consultations, one to help plan the research (including the focus areas and key populations) and another to discuss and validate preliminary results (a third, to validate final results, is pending). These consultations included the leaders of informal mining associations as well as government ministries of commerce, mining, and health and local NGOs/UN organizations (e.g., UNICEF). Overall, these revealed that there is in practice virtually no regulation of these markets by government or civil society: while some regulations may exist on paper, they are essentially never enforced in these remote, fast-changing markets that serve marginalized consumers in remote areas. That said, government officials do have concerns related to these markets, particularly with regards to food safety and the sale of expired or adulterated products—but they do not have the resources to act on those concerns.
- We have revised the text to note both the existence of such consultations, as well as the other methods (p. 3, lines 134-140), and the fact that government is not actively involved in regulating these markets (p. 3-4, lines 165-168). The latter point is brought up again where relevant in the discussion section (p. 12, lines 382-383).
R3_Comment04. A study like this where data is collected from potentially vulnerable individuals should have been approved by the relevant research ethics committee at one of the authors institutions and this should be mentioned. If there was no ethics approval for the method, then how do I know that it was approved and done to a known standard. Not highlighting the ethics approval threatens the reputation of the journal.
- Thanks for the reminder. Ethical clearance has been included in the paper line 142-144. This study was approved by the Comité National d’Ethique pour la Recherche en Santé (CNERS) in Conakry, Republic of Guinea on July 6, 2018 (N: 080/CNERS/18).
R3_Comment05. I assume that in an informal mining settlement many different cultures are brought together so a lot of different food traditions might be present. Not sure if this is covered or even asked.
- What we saw in the cross-sectional survey conducted for the larger research study was that a significant proportion of the miners interviewed (~33% in Wave 1, 42% in wave 2) were local to the zone. Additionally, while there might be many food cultures represented by the individual miners, their ability to recreate those cultures are often limited by what is actually available to purchase and use. R.e. staple crops - rice is the predominate crop, and is more widely available than other staples, such as maize (although people can find it and do eat it). We had a couple participants in the in-depth interviews explain that they were originally from Southeastern Guinea (Guinea forestiere), where pork was much more commonly eaten - however they were unable to eat it in the mining communities because the majority of the population was Muslim, and wouldn't serve it or keep pigs for sale and eating. It would appear, at least from these initial results, that people bring what they can, but are largely limited by what is actually available in the camps.
R3_Comment06. p2 line 62. Are there no positive outcomes? What about opportunities for women to enter the food service business. it would be good to posit some benefits as well as many down-side issues.
- One could argue that the existence of mining camps offers new business opportunities for women for selling food. As noted in Table 2 and lines 205-209, all of the vendors we interviewed (and all the food vendors we encountered at the mining sites for that matter) were women, and eight of the twenty vendors came from villages that had no mining activity. However, we must point out that the market is quickly saturated once word gets out that a mine is productive - our interviews with food vendors showed that there wasn't a lot of room for expansion or market diversification - everyone pretty much sold the same types of items and didn't really see the incentives in selling a new or untried item.
R3_Comment07. p2 line 64. It is essential. Maybe use a more passive voice?
- I agree that wording may be too strong here. The language has been modified to be more passive, using “it would be reasonable to…”.
R3_Comment08. p2 para ending line 68. I have the feeling that formal mining has received a lot of research interest - for example in South Africa and Latin America. Some more reference to this would have been helpful.
- You are correct that the topics of nutrition and food choice among formal-sector miners has received some research attention, primarily with a focus on dietary intake. We have added reference to this in the text (p. 2, lines 48-52). (Certainly, there has been extensive research on miners and other public health issues (e.g., lung diseases, occupational injuries), but that is not very relevant to the scope of this paper and is not mentioned here.)
R3_Comment09. p2, line 82. This is a very narrow definition of market research, so you need to specify that it is specific to this research.
- Thanks for the suggestion. The specification has been made on line 87.
R3_Comment10. p2, from line 90. How were these questions selected?
- This study was one in a series of studies funded through the Drivers of Food Choice competitive grants program, funded by the Bill and Melinda Gates Foundation and administered by the University of South Carolina. These are drawn from the larger research agenda on drivers of food choice developed by the program. The question about intervention responds to concerns raised by local stakeholders when the project was launched in Kankan Region.
R3_Comment11. I feel that this section would have benefited from a conceptual framework, even if only a simple one.
- Thank you for noting this. The conceptual framework for this study is the 5 dimensions of ‘‘food access’’ (availability, accessibility, affordability, accommodation, acceptability. This framework is supported by the systematic review “The local food environment and diet: A systematic review” by Caspi et al. 2012, which in turn was built on the conceptual definition proposed by Penchansky and Thomas (1981). We have included this information in the “Analysis” subsection of “Materials and Methods.”
R3_Comment12. p3, line 96. You may have demonstrated complexity, but was this in answer to a research question, or just an incidental outcome. Vendors responding to customers in a competitive environment is not a particularly exciting or surprising finding.
- We agree that “complexity” per se is not a novel finding, but it is a finding nevertheless. We interacted with the mining sites over a period of 18 months, and observed constant changes in location and arrangement of markets, as well as mining sites being abandoned and new ones created. At the same time, we also observed some change in the products for sale over time. Therefore we contend that the term complexity is appropriate.
R3_Comment13. p3, line 117 on. This is where ethical clearance needs to be mentioned.
- As noted above, the ethical clearance has been included in lines 142-145.
R3_Comment14. p4. When thinking about the method, I was left wondering why a counterfactual had not been included - ie., a community with no mining activity going on. This would have given a control for the other cases.
- A counterfactual is a logical suggestion. However, the existence of markets at sites of artisanal mining in remote rural areas is a response to the demand from the miners for food, and their access to cash income. Markets are not found in remote rural areas where the population is composed primarily of subsistence farmers with limited access to cash income. In such situations, markets are found only in larger towns. Therefore, it would be difficult to identify a suitable counterfactual.
R3_Comment15. Figs 1 - 3. The key is the same for all pictures, but not everything appears in all figures - for example covered area. I think this needs to be adjusted. Do these work if smaller? They take a lot of space.
- Thanks for these suggestions. We have modified the figures accordingly and made them smaller.
R3_Comment16. Table 3. It is not clear how this table was made. Is this the opinion of the authors or was a process of some kind? It contains some value judgements e.g., about food hygiene. I think the source needs to be authors opinion.
- Table 3 summarizes our findings from the study, based on analysis of the data presented elsewhere in this manuscript, as well as seasonal quantitative market surveys, which are presented elsewhere [24].
R3_Comment17. Throughout the piece ethnography seems to be absent. What about traditional and cultural practices? Beliefs and taboos related to food etc? Many food and dishes in a country such as Guinea come associated with beliefs and practices related to culture.
- Interesting ideas you bring up. Unfortunately, it was not possible to present all available data in this one paper. We have made the decision to present ethnographic data from this study on households and their interaction with markets in considerable detail in a separate paper [25]. Given that this component on markets was an exploratory piece of research, we did not design the data collection to include ethnographic methods.
R3_Comment18. There is no mention of forest foods or wild crafted foods or herbal remedies in table 4. I do not believe that a mine in the middle of a forest does not have somebody selling stuff that has been wild harvested. I also do not believe that none of these markets had no green leafy vegetables or tomatoes. Because none of the supply chains/ value chain have been unpacked, we do not know where all this food really came from or which other actors are 'hidden' from view. We also discover very little about the vendors themselves, which is a bit of a shame.
- I agree it would have been intriguing to trace where all the food really came from. Based on our interviews with vendors and market authorities, most of the foods were transported by vendors from the larger cities in Guinea, namely Kankan and Siguiri. It should be noted that there were in fact green leafy vegetables (sweet potato leaves) at most markets as well as tomatoes at the three central village markets, as noted in Table 4.
R3_Comment19. p11, line 266. Some terms are used very vaguely in this paper. An example in this line is 'spoiled'. What does this mean. Often food that is damaged or starting to physically deteriorate still has a value, particularly for the poorest.
- Thanks for this feedback. Terms such as “spoiled,” “dirty,” and “expired” were specifically used by interview participants and translated by the local interviewers when prompted about what foods they perceived as healthy or not healthy. It would have been helpful if we asked for more details to clarify exactly what they meant by these terms, and it is rather unfortunate that we did not do so at the time.
R3_Comment20. The piece is strangely silent about the role of any government actors in managing these vending sites. Maybe there is no role.
- As noted above, a series of stakeholder consultations conducted before and during the project indicated that government actors play essentially no role in managing these market sites. This is noted in the revised manuscript (p. 3-4, lines 165-168), including where relevant in the discussion section (p. 12, lines 382-383).
R3_Comment21. p12, line 296 onwards. I don't see how you can detach the food environment from the other needs of miners. Other studies show that mines attract alcohol sellers, prostitutes, drugs, etc etc. Miners need equipment as well as food.
- Thank you for this insightful comment. This study had a focus on drivers of food choice. We did not systematically investigate alcohol consumption, drug use and commercial sex work. These topics were outside of the scope of this study, but we agree that evaluating them would be important for a holistic assessment of artisanal miners and their lives.
R3_Comment22. The idea of accommodation in introduced here, but there is no mention of this in the introduction. If you were going to test whether accommodation is present, then this should have been a hypothesis and it would have improved the piece.
- The existence of accommodation was not an original hypothesis of the research. However, it proved to be an emergent finding from the research, based on the responses of the food vendors interviewed, who mentioned their strategies for accommodating consumers’ needs. As such, we feel it is better to include it within the Discussion section, which reflects on the results of the research (both expected and not) as opposed to in the introduction, which is more focused on the original motivation for the research.
R3_Comment23. p12, line 334. I don't think twigs is the right term - should be stick or poles.
- Thanks for pointing this out – it has been changed to sticks in the paper.
R3_Comment24. line 337. I find the recommendation about non-staple food vendors a bit of a stretch from the data.
- Based on our observations, most of the meals prepared by vendors at the mining sites were high in staple foods (spaghetti, rice, sandwiches served on baguettes). Non-staple food vendors were largely absent.
R3_Comment25. p13, line 352. They correctly cite Turners advice, but I see no evidence that they followed it, ie., characterised or classified markets, at least not adequately.
- Thanks for pointing out this concern. We did not mean to suggest that we have thoroughly characterized and classified markets in this paper. Rather, given the complexity of markets we found here, we mean to emphasize Turner’s advice that there is a need to characterize and classify the markets. We have reworded line 417 to clarify this point.
R3_Comment26. line 364. I am very skeptical about this finding, particularly as they have not included analysis of the various different actor’s views on the issue because they did not interview policy makers, donors or government actors - so this is just opinion.
- Thank you for your feedback. As noted earlier, this study is informed by the key informant interviews and stakeholder consultations with government ministries and local NGOs/UN organizations, which revealed little to no regulation of these markets by government or civil society (lines 165-168).
Reviewer 4 Report
Reviewer comments
Manuscript ID: foods-732759
With this paper, the authors characterize eight markets that serve people engaged in artisanal and small-scale gold mining (ASM) in the rural Upper region of the Republic of Guinea, West Africa. The aim of this study was to understand how these markets worked and if they had any concerns with nutritional outcomes and food security.
In my opinion, the manuscript is not suitable for the publication in Foods, because this manuscript has several problems, especially in the work planning and in the data treatment. This whole situation results in a lack of results. Because of this, the authors cannot take consolidated conclusions.
Author Response
R4_Comment01. With this paper, the authors characterize eight markets that serve people engaged in artisanal and small-scale gold mining (ASM) in the rural Upper region of the Republic of Guinea, West Africa. The aim of this study was to understand how these markets worked and if they had any concerns with nutritional outcomes and food security.
In my opinion, the manuscript is not suitable for the publication in Foods, because this manuscript has several problems, especially in the work planning and in the data treatment. This whole situation results in a lack of results. Because of this, the authors cannot take consolidated conclusions.
- Thank you for this feedback. We would be happy to respond in greater detail and revise our manuscript accordingly, if the reviewer could identify specific issues that are of concern with the manuscript. With regards to work planning and data treatment, we have added a sub-section detailing our rationale behind choosing our specific study sites (lines 105-132). Additionally, we described the larger context in which our research study took place, including stakeholder consultations and informant interviews (lines 134-140). We added in the ethics statement (lines 142-145). Furthermore, we elaborated upon the type of analysis that took place (lines 170-188) in a new sub-section. We hope that these revisions will be useful in demonstrating how we reached our results and conclusions.
Round 2
Reviewer 4 Report
The authors have provided a new and improved version of the paper.
Final comments and considerations: In my opinion, the manuscript is suitable for the publication in Foods.